# Water Body Extraction in Remote Sensing Imagery Using Domain Adaptation-Based Network Embedding Selective Self-Attention and Multi-Scale Feature Fusion

**Jiahang Liu * and Yue Wang** 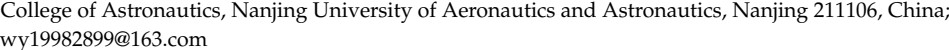

College of Astronautics, Nanjing University of Aeronautics and Astronautics, Nanjing 211106, China; wy19982899@163.com

* Correspondence: jhliu@nuaa.edu.cn or jhliu003@gmail.com

**Abstract:** A water body is a common object in remote sensing images and high-quality water body extraction is important for some further applications. With the development of deep learning (DL) in recent years, semantic segmentation technology based on deep convolution neural network (DCNN) brings a new way for automatic and high-quality body extraction from remote sensing images. Although several methods have been proposed, there exist two major problems in water body extraction, especially for high resolution remote sensing images. One is that it is difficult to effectively detect both large and small water bodies simultaneously and accurately predict the edge position of water bodies with DCNN-based methods, and the other is that DL methods need a large number of labeled samples which are often insufficient in practical application. In this paper, a novel SFnet-DA network based on the domain adaptation (DA) embedding selective self-attention (SSA) mechanism and multi-scale feature fusion (MFF) module is proposed to deal with these problems. Specially, the SSA mechanism is used to increase or decrease the space detail and semantic information, respectively, in the bottom-up branches of the network by selective feature enhancement, thus it can improve the detection capability of water bodies with drastic scale change and can prevent the prediction from being affected by other factors, such as roads and green algae. Furthermore, the MFF module is used to accurately acquire edge information by changing the number of the channel of advanced feature branches with a unique fusion method. To skip the labeling work, SFnet-DA reduces the difference in feature distribution between labeled and unlabeled datasets by building an adversarial relationship between the feature extractor and the domain classifier, so that the trained parameters of the labeled datasets can be directly used to predict the unlabeled images. Experimental results demonstrate that the proposed SFnet-DA has better performance on water body segmentation than state-of-the-art methods.

**Keywords:** water body extraction; remote sensing images; selective self-attention module; multi-scale feature fusion module; domain adaptation

## 1. Introduction

As an essential element of the earth life support system, water is crucial for sustainable development [1]. The extraction of water body information is of great significance to many fields, such as flood monitoring, military reconnaissance, wetland protection, and cartography [2–4]. With the development of remote sensing, the acquisition of water body information from remote sensing images has become the main approach with high efficiency and low cost. However, it is still a challenging task to accurately extract water bodies from remote sensing images. On the one hand, water bodies usually show different appearances due to the effect of many factors, such as green algae and sediments. Moreover, there are many types of water bodies including lakes, ponds, rivers with many small branches, which leads to the great difference in size and shape. As a result, it is difficult to identify all the water bodies simultaneously. On the other hand, the edge of the water body is

irregular by the influence of surrounding shadows, soil, vegetation, etc., making the water edge detection more complex than buildings and roads [5]. How to accurately extract a water body from remote sensing images, especially high-resolution images, is one of the hot topics in recent years.

In the past decades, many water body detection methods have been published in the literature, and they can be roughly categorized into water index methods, traditional machine learning algorithms and deep learning algorithms. Water index methods are simple and widely used; they rely on manual characteristics, such as the bands ratio. The normalized difference water index (NDWI) is one of the most popular methods [6,7]. It uses the characteristic that the water body absorbs more energy in the NIR band than in the green band to calculate the water index and extract water bodies in which NDWI is bigger than a certain threshold [8]. As NDWI relies too much on spectral information and ignores spatial characteristics, it is difficult to determine the threshold when there are other categories of objects, such as buildings, shadows, and roads, in the images [9]. Because the similar reflection patterns of water and shadow would lead to misclassification, Feyisa et al. [10] designed the automated water extraction index (AWEI) to increase the gap between water and non-water bodies, and the AWEI has good performance on the Landsat dataset. Fisher et al. [11] proposed $WI_{2015}$ based on $WI_{2006}$ and compared it with NDWI and AWEI. The experimental results show that NDWI has high commission errors in quarry, soil, and urban areas, while $WI_{2015}$ and AWEI have good accuracies, especially for cloud-shadow pixels. However, the AWEI and $WI_{2015}$ do not produce unified results for all kinds of data [12], and the AWEI is confined to five case studies and one step [13].To meet the requirements of automation, researchers try to use some machine learning algorithms to automatically extract water bodies from remote sensing images [14], such as pure Bayesian (NB), decision tree (DT), and support vector machine (SVM). Yao et al. [7] proposed an automatic water body extraction method combining SVM with NDWI. In the method, SVM is used to determine the optimal coefficient of the water index and generate a dark building shadow removal model. As the building and shadow do not always satisfy the proposed geometric relationship, the segmentation accuracy of this method may be greatly impacted [15]. At the same time, considering that the water surface may be covered by phytoplankton or aquatic vegetation, it is impossible to exactly detect water bodies [7]. Khandelwal et al. [13] created a new classification model by using seven reflectance bands of two MODIS products with 500 m resolution and SVM classification model and combined the model with the noise correction method, which shows a good effect. Tri et al. [16] used J48 decision tree (JDT) to identify water body in images. Although its segmentation accuracy is fine, it requires professional knowledge as well as strict parameter and data selection. More importantly, similar to the water index methods, the traditional machine learning algorithms cannot acquire the context information of the images.

In recent years, the convolutional neural network (CNN) has shown strong feature expression ability and has been widely used in the field of remote sensing [17,18]. Yu et al. [19] introduced CNN to water body detection in remote sensing images. However, due to the shallow network structure, it is insufficient in learning ability and robustness. Because the shallow layers are generally universal filters and the wetland species in Canada are similar, Rezaee et al. [20] used the pre-trained model to fine-tune the parameters rather than complete training to predict the RapidEye imagery. Different from the methods of putting image slices into CNN, a fully convolutional network (FCN) abandons traditional fully connected layers and utilizes convolutional networks to classify an entire image at pixel level [21]. For this property, FCN not only can meet the condition of different image sizes, but greatly improves the efficiency and precision of the segmentation. Isikdogan et al. [22] proposed the FCN-based model, named DeepWaterMap, which changes the number of model parameters and the way of layer splicing, making the network more suitable for water segmentation. Li et al. proposed a method based on FCN to extract water information from high-resolution remote sensing images [15]. Compared with other methods mentioned above, the FCN can obtain better results, but the edges of the final

predicted water bodies are always blurred. What is more, small water bodies are often missed as lots of feature in the process of continuous downsampling, which expands the receptive fields. To overcome this deficiency, U-Net for biomedical image segmentation is presented [23]. The U-Net employs a symmetrical encoder–decoder structure, which is linked by skip connections to settle the problem of resolution degradation. Feng et al. used Deep U-Net and a superpixel-based conditional random field (CRF) model to detect water bodies [24]. It has poor performance on small water areas, as it directly concatenates different levels of feature maps and makes the characteristics confusing [25]. Inspired by Resnet's ability to continuously reuse information and Densenet's ability to acquire new features, Shamsolmoali et al. [26] combined Resnet and Densenet to achieve better segmentation while being GPU-friendly. Multi-scale water extraction convolutional neural network (MWEN) [9] adds a multi-scale dilated convolution in the last layer to combine the information in different scales, but it is insufficient since the multi-scale extractor is only added on the last layer. Considering these aspects, a multi-feature extraction and combination network (MECNet) [5] is proposed to integrate different receptive fields and channel information. However, MECNet only considers global information and ignores spatial information, which leads to the detail loss. What is more, the several methods mentioned above are based on the structure of the encoder–decoder, though this structure alleviates the problem of information loss during the downsampling process, the low-level feature maps also lose unrecoverable details that are vital for the prediction. Different from the traditional encoder–decoder architecture, a novel network HRnet V2 [27] is proposed. In this network, four parallel branches are used to acquire and fuse features, and the first branch maintains a full resolution image. However, if HRnet V2 is directly applied to water segmentation without considering the characteristics of water bodies, it still leads to location information loss. Additionally, it is known that high accuracy lies in enough labeled samples for training almost all DL methods. However, due to the lack of labeled samples in real applications, the deep learning methods are difficult to achieve ideal results in practical remote sensing application. At same time, the mount of labeling samples is costly. The multiscale residual network (MSRnet) obtains the semantic information of the images through self-supervised learning strategies and predicts unlabeled images through the combination of semantic information and supervised learning parameters [28]. Although the MSRnet has great performance, it needs labeled samples, as it directly shares the parameters of supervised learning. Among all the above methods, the water index methods lack automatic detection abilities, and the machine learning methods rely on feature and data selections. Although the water detection accuracy produced by DL is improved, the DL methods cannot obtain semantic and location information simultaneously and require the mount of labeled samples, which indicates that although there are many excellent water segmentation studies, there exists room for progress.

To overcome these defects mentioned above, a selective feature enhancement and multi-scale feature fusion network based on domain adaptation (SFnet-DA) is proposed in this work. To improve the ability of anti-noise and identification under the premise of ensuring the location information, the SFnet develops a spatial information-based selective self-attention (SSA) mechanism that employs different feature enhancement methods for various branch fusions, which selectively increases the details or semantic information of the images, making the top-down branches hold a state of increasing semantic information and decreasing location information. With SSA structure, the network can resist the noise disturbances, such as clouds and green algae, while having the ability to identify small and vast water bodies simultaneously. A multi-scale feature fusion (MFF) module is designed to enhance the location and detailed information of the feature map without increasing the number of network parameters, which raises the accuracy and connectivity of the water edge. Furthermore, to predict more refined images, the fully connected CRFs [29] are applied to post-process the predicted images. To avoid image labeling, the domain adaptation, which utilizes the existing labeled dataset to predict the target unlabeled dataset, is preferred. The feature distribution of the labeled dataset is related to but likely different

from the feature distribution of the target dataset. The labeled dataset is defined as the source domain, and the target dataset is defined as the target domain. The SFnet-DA builds an adversarial relationship between the feature extractor and domain classifier to make the feature distribution of source domain and target domain consistent, namely domain invariance. Specifically, the feature extractor maps the images of the source domain and the target domain so that the domain classifier determines that the features from the source domain and the target domain are source domain features as much as possible, and the domain classifier correctly determines whether the features come from the source domain or the target domain as much as possible. In addition, to ensure the prediction effect of unlabeled samples, we connect the feature extractor with SFnet to train through source domain images and labels. After learning, the mapped features from –source domain and target domain are both domain invariant and discriminative, so the target domain can predict the unlabeled samples by using the parameters trained in the source domain.

The main contributions of this paper can be summarized as the following.

(1) We designed a novel SFnet composed of SSA, MFF, and fully connected CRF post-processing modules. Specifically, SSA selectively intensifies the features of different scales and enhances the capability of anti-noise. The combination of fully connected CRFs and MFF, which changes the proportion of each branch feature, and the splicing method not only can improve the position information of feature maps, but can optimize the edges of water bodies.

(2) To reduce the demand for labeled samples, the adversarial domain adaptation method, for the first time, is embedded in the proposed SFnet-DA. Specifically, through the adversarial relationship between the feature extractor and the domain classifier, the labeled dataset and the unlabeled dataset are connected, which enables us to directly use the parameters trained by the labeled dataset to predict the unlabeled images.

(3) The designed water detection method combines SFnet with adversarial domain adaptation (SFnet-DA), and the test results of our method on public datasets are better than other advanced methods.

The rest of the paper is structured as follows. Section 2 introduces the methods, including the design of SFnet, the domain adaptation method, and fully connected CRFs. The evaluation data, experimental settings, algorithm results, and discussion are shown in Sections 3 and 4. Finally, the summarization is given in Section 5.

## 2. The Proposed Method

With the domain adaptation (DA) mechanism, the proposed SFnet-DA has four branches. The lower branches of SFnet are responsible for the location and detailed information of the water bodies, and the upper branches of SFnet are responsible for the semantic information. To obtain the water body information of different sizes and alleviate the influence of noises, the SSA is embedded into SFnet. To obtain an accurate water edge, SFnet embeds the MFF module and takes the fully connected CRFs as the post-process method. Then the combination of SFnet and DA is used to construct SFnet-DA to avoid labeling tasks. Figure 1 is the flowchart of the proposed method. First, we mark the source domain images as 0 and the target domain images as 1, and put them into the feature extractor to generate the feature vectors, which would enter the domain classifier. The feature extractor maps the images of the source and target domains, and the domain classifier determines which domain the mapped feature vectors come from, so that the resulting adversarial relationship makes the features mapped in the source and target domains similar and achieves domain invariance of the feature vectors. Then, the feature vectors of the source domain images generated by feature extractor will be processed by SFnet. The SFnet trains parameters together with the source domain labels, which guarantees the discriminability of the mapped feature vectors, i.e., the water segmentation performance of the network. After training, the mapped features from source domain and target domain are both discriminative and domain invariant, so the target domain can directly use the SFnet, trained in the source domain, to predict its own unlabeled samples.

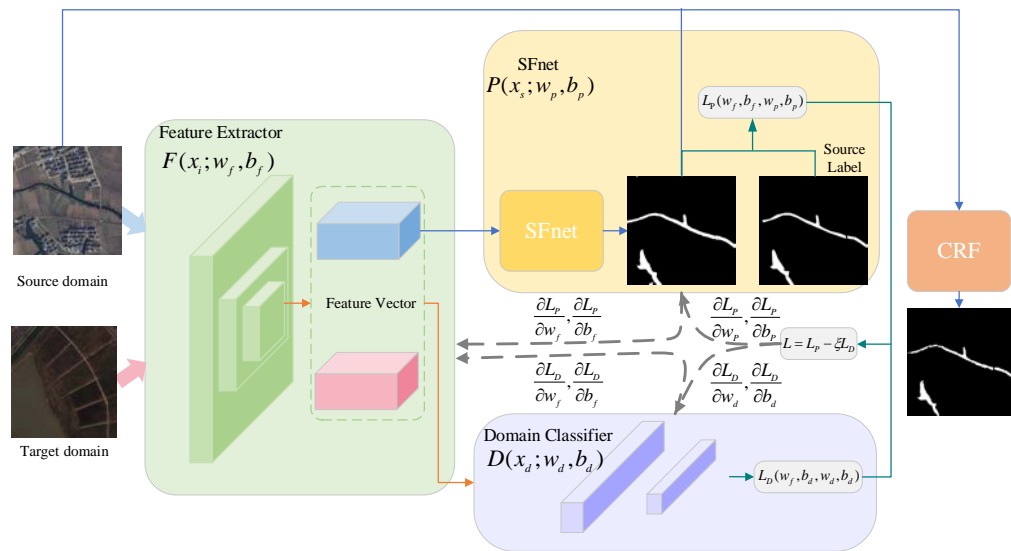

**Figure 1.** The flow chart of the SFnet-DA.

### 2.1. Structure of SFnet

HRnet V2 is a high-resolution neural network for human pose estimation, and it is used as a basic framework for SFnet. The structure of the SFnet is shown in Figure 2. The SFnet is composed of four branches. The first branch is used to extract features in the full resolution of input images for ensuring the spatial position of the image objects. The feature maps of the other three branches are 1/4 of the characteristic graphs of each previous branch for capturing more context information. To ensure that the entire network is performed with a small amount of calculation, the first feature extraction module of the first branch is different from other modules. Specifically, it consists of two dilated convolutions, the rectified linear unit (ReLU) activation function, the batch normalization (BN) layer, and the BottleNeck. The BasicBlock used in other feature extraction branches is to deepen the network and extract the feature map. At the same time, it can catch the count of feature channels and image resolution without change. For the purpose of expressing characteristics better, the downsampling module is a dilated convolution whose output channels are twice that of input channels.

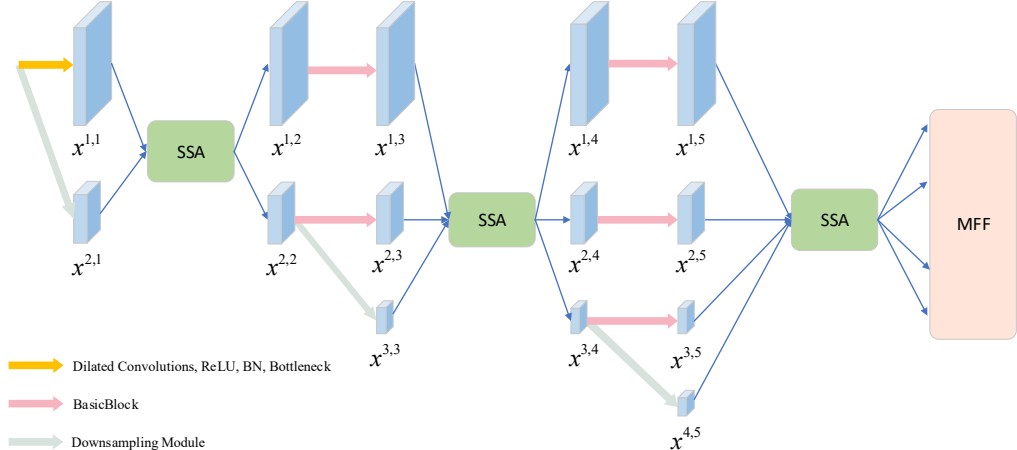

**Figure 2.** SFnet structure diagram. $x^{m,n}$ represents the feature map processed by the BottleNeck, BasicBlock or downsampling modules. The paired images at the same stage enter the SSA, and the features after the last layer of SSA are input into the MFF for feature fusion.

Let $x^{m,n}$ represent the feature map after the BottleNeck, BasicBlock or downsampling modules, where $m$ denotes which branch the feature map is located in, and $n$ indexes

the feature extraction stage. The features located in different branches at the same stage enter the SSA for integration among them. In the relatively primary branch, the SSA prefers to enhance the edge and detail information of the advanced branch by training a self-attention parameter and guides the high-level feature to adaptatively discover the salient area of each image. Similarly, in the relatively advanced branch, the SSA prefers to enhance the ability to detect water bodies of different sizes by helping the primary branch make full use of the large receptive field information of a high-level map. After the last SSA operation, the network gains four-level feature maps with gradually decreasing resolution and increasing channel numbers as well as semantic information. The feature graphs in the low feature branches hold more detailed information but have lower semantic information and larger noise, while the feature graphs in the high-level feature branches have more semantic information and poor perception of details. To make use of their advantages, a multi-scale feature fusion (MFF) module is used to effectively combine low-level features with high-level features so that the image location information and semantic information can be assured. The SSA and MFF are introduced in detail in the following.

### 2.1.1. Selective Self-Attention (SSA) Mechanism

Since the first branch retains the resolution and the other branches expand the receptive field progressively, the functions of the four branches are different. The relatively primary feature branches are responsible for guaranteeing the contour accuracy and connectivity of water bodies. The advanced feature branches after downsampling, possessing larger receptive fields and feature channels, can resist the interference of noise and effectively identify water bodies of different size. To make different branches have different functions, we propose a simple but effective selective self-attention (SSA) mechanism which automatically adjusts the self-attention coefficient according to the feature branch number. By the self-attention coefficient, the SSA uses the characteristic of the feature branch to weight the characteristics of another branch in the feature reuse phase to realize the feature magnification.

The structure of the SSA is shown in Figure 3. Figure 3A shows the shallow feature guiding the deep feature branch and Figure 3B shows the deep feature map guiding the shallow feature branch. In Equations (1) and (2), we pick out the feature map of the first stage of the first branch $x^{1,1}$ and the map of the $n$-th phase of the $m$-th branch $x^{m,n}$:

$$x^{1,1} \in \mathbb{R}^{h \times w \times c} \tag{1}$$

$$x^{m,n} \in \mathbb{R}^{\frac{h}{2^{m-1}} \times \frac{w}{2^{m-1}} \times 2^{m-1}c} \tag{2}$$

where $h$ and $w$ are the height and width of the feature map in the first stage of the first branch, respectively. The resolution of the other branch is $4^{-(m-1)}$ of the original resolution. The counting of characteristic channels in the first branch is stated by $C$, and the number of characteristic channels of $m$-th branch is $2^{m-1}C$.

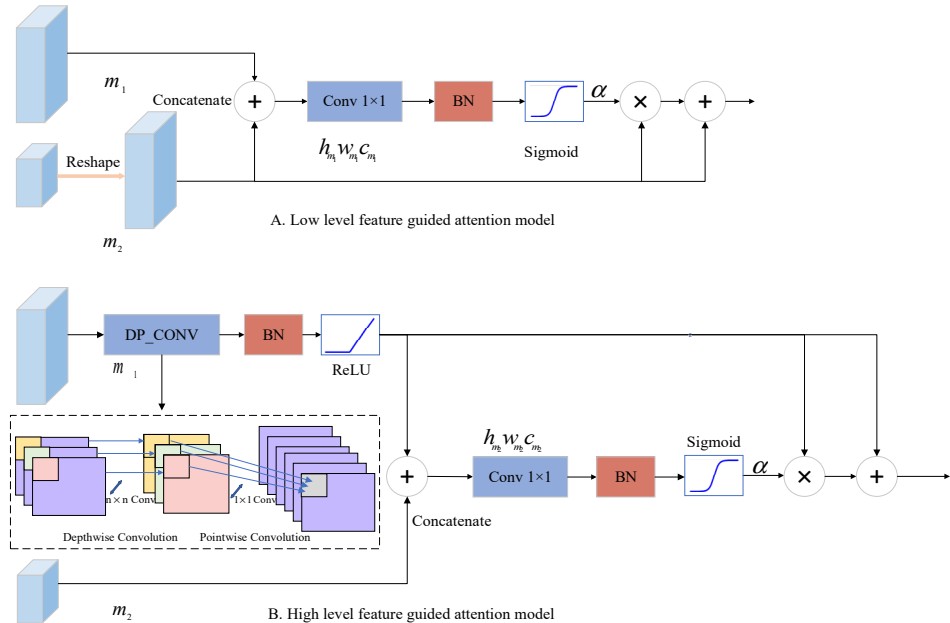

**Figure 3.** The structure of SSA. The low-level feature guided attention module and the high-level feature guided module are illustrated in (**A,B**).

To identify the salient areas of the images and choose to enhance the details or semantic information of the water body, SSA employs a differentiable soft attention mechanism. The self-attention coefficient $\alpha$ guided by a low level feature would be automatically updated, and the importance of each pixel is adaptatively determined when the network is backpropagated, so it is easy to identify the salient areas of the images and keep the details of water bodies. The $\alpha$ guided by the deep feature conducts the shallow feature to enhance the semantic information of the primary branch, so that the network would boost the capacity to distinguish petty and vast water bodies, and not be influenced by noise, such as roads and shadows. No matter what kind of guidance SSA adopts, the sum of low-level and high-level features is used to generate the self-attention coefficient $\alpha$, which considers the global information of the feature maps. Instead of choosing a generalized global average pooling, the combination of $1 \times 1$ convolution and BN layer is used to acquire a grid signal based on position information, which is conducive to interacting and integrating different channel information. What is more, the SSA is lightweight, since the output channel number is 1. The sigmoid activation function is used to obtain a convergent coefficient since the consequent use of softmax results in a sparse output.

$$X_{att} = B(W^T(X_i + x_i^{re})) + b_{att} \tag{3}$$

$$\alpha = \frac{1}{1 + \exp(X_{att})} \tag{4}$$

$$X_1 = (\alpha + 1)X \tag{5}$$

where $B$ denotes the batch normalization (BN) layer, and $W$ and $b_{att}$ denote the weight and bias, respectively. $x_i^{re}$ denotes the map which retains its own branch, $X_i$ denotes the map which is processed by the network to get the same size with $x_i^{re}$, and $X_1$ is the output of the SSA mechanism. For the case of selecting the primary characteristic branch to be reserved in Figure 3A, $X = U(x^{\max(m_1, m_2,),n})$ and $x^{re} = x^{\min(m_1, m_2,),n}$, where $U$ denotes the upsampling process. In Figure 3B, when the feature processed by SSA is retained in

the advanced feature branch, depthwise separable convolution (DP_CONV) is utilized to lessen the storage of the SSA, namely

$$X = RB(x_i^{\min(m_1, m_2), n} * W_{j*c+i}) \tag{6}$$

$$x^{re} = x^{\max(m_1, m_2), n} \tag{7}$$

where $W_{j*c+i}$ denotes the weight of the depthwise convolution with kernel size n × n and the pointwise convolution with kernel size 1 × 1. $i \in [1, c], j \in [0, m-1]$, $m$ represents channel expansion and $RB$ represents the ReLU activation function combined with the BN layer.

To maintain the original characteristics of the branch and let the branch with a larger $m$ contain more semantic information, we add all feature maps in the same branch after passing through the SSA mechanism, and then add the sum with the feature $x^{m,n}$ of the branch itself:

$$x^{m,n+1} = \sum_{i=1}^{K, i \neq m} A(x^{i,n}) + x^{m,n} \tag{8}$$

where $K$ is the maximum number of branches, and $A(\cdot)$ is used to represent the SSA mechanism whether the SSA is located in the low-level feature branch or the high-level feature branch.

### 2.1.2. Muti-Scale Feature Fusion (MFF) Module

Generally, if the deep network is simply connected at the end, the details of the feature maps would be lost, and the edges of the water bodies would be blurred. To better maintain the detailed features, we design a light multi-scale feature fusion (MFF) module which integrates the location and semantic information while avoiding the network overfitting. To the proposed SFnet, a basic purpose is that the primary features of the images are extracted by shallower branches, and the advanced characteristics of the graph are perceived by deeper branches. In the downsampling process that expands the category expression ability of the images, the number of feature channels of the branch is changed to $2^{m-1}C$. After the last SSA mechanism and adding process, we obtain four-level features and the high-ranking feature channels become $8C$; simple splicing would inevitably pay much attention to the rich semantic information of the deep feature map and neglect the basic characteristics. However, for the water segmentation of high-resolution remote sensing images, the accurate location of the water body is instrumental in the correct positioning and area calculation. Hence, we use a standard convolution module $NCONV^m$ including the 1 × 1 convolution layer, BN layer and ReLU activation function to reduce the number of feature channels from $2^{m-1}C$ to $C$, which increases the proportion of low-level feature branches while reducing the model parameters.

What is more, compared with the direct splicing, 'stepped' concatenating is adopted. That is, the advanced feature branch is not directly upsampled to the first branch after the standard convolution module, but only upsampled to four times that of the advanced feature map and concatenated with the previous images after the standard convolution module. The structure of MFF is shown in Figure 4 and the overall process is as follows:

$$NCONV^m = RB(W^T x_i^{m,n} + b) \tag{9}$$

$$S^m = \begin{cases} U(NCONV^4) \oplus NCONV^3 & m = 3 \\ U(S^{m+1}) \oplus NCONV^m & \text{otherwise} \end{cases} \tag{10}$$

where $\oplus$ is the dimension concatenating, $S^1$ is the output of the MFF, and $NCONV^m$ is the standard convolution module of the m branch, which includes the 1 × 1 convolution layer, BN layer and RELU activation function. $W \in \mathbb{R}^{C' \times C}$, $C' = 2^{m-1}C$ and $b \in \mathbb{R}^C$. We use this structure to lighten the network and improve the accuracy of the water detection.

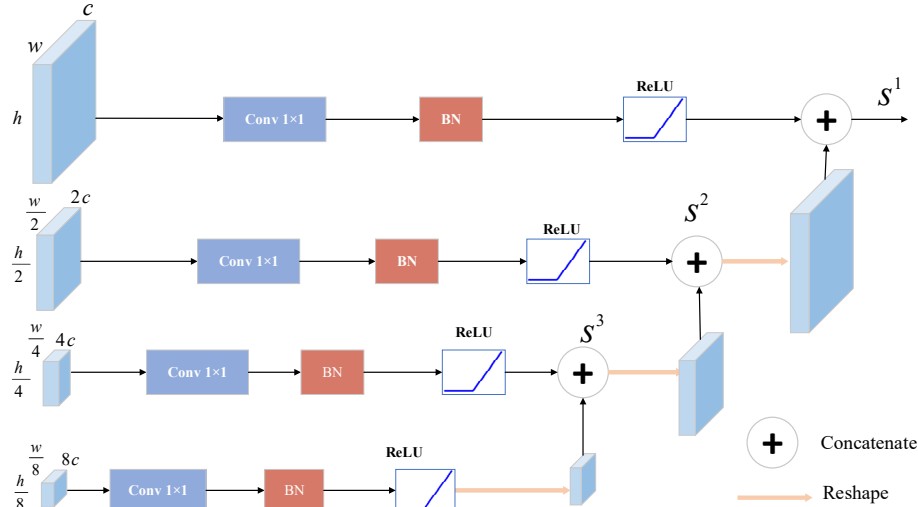

**Figure 4.** The structure of MFF. *h*, *w* and *c* are respectively the height, width and characteristic channels numbers of the first stage characteristic graph of the first branch. The four-level features in the figure are the output after SSA and its adding process.

### 2.2. SFnet Based on Domain Adaptation (SFnet-DA)

Generally, the water detection methods based on deep convolution neural network utilize a large array of labeled samples for training; however, manual labeling is extremely time consuming and expensive. To skip labeling, we attempt to use the existing labeled dataset to predict the unlabeled target dataset. However, the model trained on one labeled dataset cannot directly predict another unlabeled dataset since there are huge differences, such as shooting area, and image resolution. Additionally, the characters of water bodies are generally different from each other in different regions, especially in high-resolution images. To the different datasets, these differences in water bodies are more common and lead to a domain shift, namely dataset bias. To address this problem, we built up our semantic segmentation network (SFnet) based on domain adaptation, which reduces the domain shift in the feature space. If the water body features of one unlabeled remote sensing imagery set need to be predicted, the domain adaptation can reduce the domain shift between unlabeled target datasets and existing labeled datasets that are highly likely to be distributed differently, making it possible to skip labeling work and directly predict unlabeled samples. The unlabeled samples are denoted as the target domain, and the labeled samples are denoted as the source domain. The input spaces of the source domain and target domain are assumed to be $X$. The annotation space of the source domain is assumed to be $Y = \{1, 2, \ldots, L\}$, where L refers to the number of categories of image segmentation. The distributions of source domain and target domain are represented by $\mathbb{S}$ and $\mathbb{T}$, respectively, and $\mathbb{S} \neq \mathbb{T}$. The parameters $x_i \in X$ is used to denote images of source and target domain, and $y_{S_i} \in Y$ denotes the labels of target domain, respectively. The parameters $n_s$ and $n_t$ are used to denote the image counts of the source domain $I_s$ and target domain $I_T$, respectively, and $I_S = \left\{x_{S_i}, y_{S_i}\right\}_{i=1}^{n_s} \sim \mathbb{S}$, $I_T = \left\{x_{T_i}\right\}_{i=1}^{n_t} \sim \mathbb{T}$. We mark all images from the source domain as 0, and all images from the target domain as 1.

To reduce the dataset bias between source domain and target domain, SFnet-DA defines the feature extractor and the domain classifier with an adversarial relationship. As the high-resolution features of remote sensing images and the large discrepancies in the scales of different water bodies, multiple convolution layers and a max pool layer are combined to define the feature extractor $F(x_i; w_f, b_f)$, it makes it possible to prevent a large loss of resolution while extracting features. $F$ maps the input images of the source domain and the target domain into the eigenvectors $f \in \mathbb{R}^{h \times w \times 3}$, which attempts to enhance the similar parts between the images of the source domain and the target domain, and reduce the dissimilar parts so that the distribution of the features in the feature space is the same,

i.e., $f_{Si} = f_{Ti}$. The domain classifier $D(\mathrm{x}_i; w_d, b_d)$ uses the domain discrimination label to determine whether the images come from the source domain or the target domain, that is, we define

$$
x_i \sim \begin{cases} \mathbb{S} & \text{if } d_i = 0 \\ \mathbb{T} & \text{if } d_i = 1 \end{cases}
\tag{11}
$$

A simple adversarial realized method, named gradient reversion [30], achieved by $\xi$, is to reverse the gradient in the backpropagation, and keep the input unchanged during forward propagation. Furthermore, to guarantee the ability of segmentation, namely discriminativeness, $F$ is connected with SFnet, whose structure does not require any changes since $f \in \mathbb{R}^{h \times w \times 3}$. The objective loss is defined as $L$:

$$
L = \sum_{i=1}^{n_s} L_P^i(w_f, b_f, w_p, b_p) - \xi \sum_{i=1}^{n_s+n_t} L_D^i(w_f, b_f, w_d, b_d)
\tag{12}
$$

$$
L_P = -\mathbb{E}_{x_{S_i} \sim X_s}(y_i \log P(x_{S_i}) + (1 - y_i)\log(1 - P(x_{S_i})))
\tag{13}
$$

$$
L_D = -\mathbb{E}_{x_{S_i} \sim X_s} \log D(x_{S_i}) + \mathbb{E}_{x_{T_i} \sim X_T} \log(1 - D(x_{T_i}))
\tag{14}
$$

where $L_P^i$ and $L_D^i$ represent the loss of the semantic segmentation and domain classifier respectively. $\xi \in R$ and $\xi > 0$. $P(x_{S_i})$ represents that SFnet trains the parameters by the source domain images and source domain labels to realize discriminability. Benefiting from gradient reversion, the adversarial relationship can reduce the dataset bias. On the one hand, $\xi$ causes the parameters $w_f$ and $b_f$ of the feature extractor to be updated to make the mapped features as consistent as possible so that the domain discriminator $D$ makes the wrong judgment, that is, all judgments are 0, and maximizes $L_D$. On the other hand, the parameters $w_d$ and $b_d$ of the domain discriminator are modified to identify where the images come from as much as possible, that is, judge the source domain image as 0 and the target domain image as 1, and minimize $L_D$. The specific back-propagation process of finding saddle points $(\widetilde{w_p}, \overline{b_p}, \widetilde{w_f}, \widetilde{b_f}, \widetilde{w_d}, \overline{b_d})$ can be expressed as Equations (15) and (16). Equation (15) is used to minimize the $L_P$ by updating the parameters $(w_p, b_p, w_f, b_f)$ for improving the water body segmentation capability of the network. At the same time, due to the gradient reversion, the $L_D$ would be maximized, and the ability of the feature extractor to make the feature vectors mapped by different domains become uniformly distributed is improved. Equation (16) updates the parameters $w_d$ and $b_d$ to minimize $L_D$, and improve the classification ability of the domain classifier:

$$
(\widetilde{w_p}, \widetilde{b_p}, \widetilde{w_f}, \widetilde{b_f}) = \arg \min_{w_p, b_p, w_f, b_f} L(w_p, b_p, w_f, b_f, \widetilde{w_d}, \widetilde{b_d})
\tag{15}
$$

$$
(\widetilde{w_d}, \widetilde{b_d}) = \arg\max_{w_d, b_d} L(\widetilde{w_p}, \widetilde{b_p}, \widetilde{w_f}, \widetilde{b_f}, w_d, b_d)
\tag{16}
$$

Therefore, the features learned by the feature extractor have both discriminability and domain invariance. When a set of optimal parameters is learned, the feature extractor almost satisfies $F(x_{S_i}; w_f, b_f) = F(x_{T_i}; w_f, b_f)$. When we want to predict the target domain images without labels, we put the target domain images $x_{T_i}$ into the feature extractor to make the $F(x_{S_i}; w_f, b_f) = F(x_{T_i}; w_f, b_f)$, and use the SFnet parameters trained with the source domain labels to segment unlabeled images.

### 2.3. Fully Connected CRFs

The performance of a network depends on enough fine samples. When the labels themselves are not fine enough, the network will lead to the predictions not being close to the original images. Additionally, owing to the restricted storage of the GPU, initial downsampling is necessary. This operation always loses the detailed information and makes the edge of water blurred and the classification of small water regions error. In recent years, conditional random fields (CRFs) were widely utilized to smooth rough

segmented images [29]. However, CRFs only use single potential energy on a pixel or paired potential energies on adjacent pixels. This property results in a sparse structure that can only couple adjacent points and implement modeling between short-distance pixels. In contrast to CRFs, each pixel in fully connected CRFs is coupled with all other pixels to form connecting edges, which greatly enlarges the quantity of paired connections and is conducive to establishing long-distance connections, restoring edges, and identifying small water bodies. Accordingly, we combine our network with fully connected CRFs.

For any conditional random field $(I, X)$, the probability distribution can be described by the Gibbs distribution:

$$P(X|\,I) = \frac{1}{Z(I)} \exp(-\sum_{q \in Q_G} \theta_q(X_q|I) \tag{17}$$

$$Z(I) = \sum_{q \in Q_G} \prod_{q \in Q_G} \theta_q(X_q|I) \tag{18}$$

where $X = \{X_1, \ldots, X_N\}$ refers to the random field, and $X_i$ is the pixel label, namely water or background. $I = \{I_1, \ldots, I_N\}$ indicates the images of input with size $N$, and $Z(I)$ is the normalization factor. $G(V, \varepsilon)$ is the complete graph on $X$, and $Q_G$ is the largest subgraph in $G$. $\theta_q$ denotes the potential function of $q$.

In fully connected CRFs, the energy of Gibbs is as follows:

$$E(x) = \sum_i \theta_u(x_i) + \sum_{i<j} \theta_p(x_i, x_j) \tag{19}$$

where the unary potential function is denoted as $\theta_u(x_i) = -\log P(x_i)$, and $P(x_i)$ indicates the probability value of the output of the SFnet. Even though $\theta_u(\cdot)$ contains the situation and color information of the water body, it is only related to the pixel point $i$, and has no association with pixels in other locations. Note that pixels with similar colors and positions are usually of the same type, so the pairwise potential function $\theta_p(x_i, x_j)$ is used to determine the probability of two events occurring at the same time:

$$\theta_p(x_i, x_j) = u(x_i, x_j) \sum_{m=1}^{K} w^{(m)} k^{(m)}(f_i, f_j) \tag{20}$$

$$k(f_i, f_j) = w^{(1)} \exp(-\frac{|P_i - P_j|}{2\tau_\alpha^2} - \frac{|I_i - I_j|}{2\tau_\beta^2}) + w^{(2)} \exp(-\frac{|P_i - P_j|^2}{2\tau_\gamma^2}) \tag{21}$$

when $i = j$, $u(x_i, x_j) = 0$; otherwise, $u(x_i, x_j) = 1$. In this definition, $u(x_i, x_j)$ would constrain the conduction condition between pixels and penalize adjacent pixels of different labels. $w^{(m)}$ denotes a combination of linear weights, $P_i$ and $P_j$ indicate position vectors, while $I_i$ and $I_j$ signify color vectors. The first appearance kernel adopts parameters $\tau_\alpha$ and $\tau_\beta$ to control the similarity and proximity while considering location and color information. The second smoothing kernel that uses the parameter $\tau_\gamma$ to control the scale of the Gaussian kernel only considers spatial information and aims to remove small isolated water bodies.

We put the original images and the water segmentation maps predicted by SFnet into fully connected CRFs post-processing module to obtain the energy of Gibbs through calculating $\theta_u(x_i)$ and $\theta_p(x_i, x_j)$, and finally obtain the optimized water segmentation map.

## 3. Experiments

### 3.1. Data

The DeepGlobe Land Cover Classification Challenge [31] and 2020 Gaofen Challenge dataset [32] are combined as the source domain of our network. Among them, the Deep-Globe Land Cover Classification Challenge obtained from the Digital Earth Vivid+ dataset is the first public dataset of high-resolution sub-meter satellite images. The dataset, including 803 labeled RGB images with a size of $2048 \times 2048$ and a resolution of 50 cm, takes

rural areas as the main object, and has seven manually labeled datasets of cities, agriculture, pastures, forests, land, water bodies, clouds, and other objects. The 2020 Gaofen Challenge dataset includes 1000 RGB images, and the image size is 492 × 492. We mix the DeepGlobe Land Cover Classification Challenge as well as the 2020 Gaofen Challenge dataset as the source domain and extract 1263 images from the dataset for training, 270 for verification, and 270 for testing. The images are resized to 512 × 512 before they are input into the network.

The Wuhan dense labeling dataset (WHDLD) [33,34] is a dataset released by Wuhan University in 2018, and is used as the target domain in this experiment. WHDLD contains 4920 RGB images that are cropped from the large remote sensing dataset of Wuhan City, and includes six manual annotations of buildings, roads, sidewalks, vegetation, bare soil, and water. The dataset with a size of 256 × 256 has a resolution of 2 m. The training part of the target domain is aligned with the source domain, and 20% and 20% of the WHDLD are selected as the validation and test set, respectively. Similarly, we resize the images to 512 × 512 before inputting.

### 3.2. Experiment Settings

For fair comparison, both our method and comparison methods were learned by the cross-entropy loss function and Adam optimizer. The initial learning rate is set to 0.0001, and the StepLR is used to adjust the learning rate to 0.92 times of it every 20 epochs. Models are written by Python and are implemented using the PyTorch framework. In addition, all experiments are carried out on NVIDIA 2060S. The batch size is set to 2 or 4, the maximum epoch is selected as 400, and the model parameters are saved in the same epoch. We compare our SFnet with U-Net, HRnet V2, DeepLab V3+ [35], DFAnet [36], and Bisenet [37]. Moreover, the SFnet-DA is validated by comparing with unsupervised methods, such as K-means [38], UIS [39], AC [40], ISODATA [41] and ISB [42].

### 3.3. Comparation

3.3.1. The Comparison of Supervised Networks

The combination of DeepGlobe Land Cover Classification Challenge and 2020 Gaofen Challenge dataset is utilized to perform supervised learning on the network. We compare it with other state-of-the-art methods, such as HRnet V2 [20], U-Net [17], DeepLab V3+ [35], DFAnet [36] and Bisenet [37], as shown in Table 1 and Figure 5, to evaluate the performance of SFnet. From Table 1, it can be found that our method is 0.82%, 0.38%, and 1.98% higher than the second-excellent algorithm DeepLab V3+ in *PA*, $F_1$, and *MIOU*, respectively. The SFnet outperforms the backbone network HRnet V2 2.21% in *PA*, 0.38% in $F_1$ and 2.24% in *MIOU*. Moreover, the *MIOU* of our network is much higher than that of Bisenet, U-Net, and DFAnet by 2.98%, 9.29% and 31.25%, which further proves the effectiveness and robustness of the SFnet. Table 1 also shows the time required for each network to train an image, and it can be found that our speed is slightly slower than that of Bisenet, DFAnet and HRnet V2, and faster than that of Deeplab V3+ and U-Net.

**Table 1.** Quantitative comparisons of models in the source domain.

| Method | PA | $F_1$ | MIOU | Time (s) |
|---|---|---|---|---|
| HRnet V2 [20] | 87.7 | 96.75 | 84 | 0.19 |
| U-NET [17] | 80.9 | 95.14 | 77 | 0.21 |
| DeepLab V3+ [35] | 89.1 | 96.75 | 85 | 0.35 |
| DFAnet [36] | 59.7 | 90.3 | 55 | 0.12 |
| Bisenet [37] | 88.4 | 96.33 | 83 | 0.14 |
| Ours | 90 | 97.13 | 87 | 0.2 |

Figure 5 shows the visual comparisons between our method and other methods. In the first row, we show the segmentation results of small rivers. It is found that other methods have some false detections, while our method can accurately detect rivers. This is

because SSA and MFF ensure that the shallow branches have detailed information, and Hrnet V2 and Bisenet, which preserve the characteristics of full resolution, have relatively less misjudgment for small information than the other three methods. Row 2–4 show the detection results when the background is complex, and the proposed method can resist the influence of green algae, green plants, roads, etc., and detect complete water body information. Specifically, the experimental results in row 2 show that except for our method, other methods are affected by grassland and green algae. In row 3, except for our method and Deeplab V3+ which uses dilated convolution, other methods do not perform well. It can be found that our results in the second and third rows perform well because the SSA module can effectively combine shallow and deep information, while other methods cannot have low-level and high-level information simultaneously. However, almost all of the images in row 4, including the images predicted by SFnet, are affected by bushes. Although U-Net and Deeeplab V3+ obtain the context information without false detection, there are some missing detections. From the last column, we can find that the image processed by the SFnet resists the influence of the bushes after fully connected CRFs post-processing. The last row shows that when there are sediments in the water, due to the SSA module and MFF module, our method is more robust, and the edge information is more accurate.

In summary, our method is superior to other networks both visually and quantitatively.

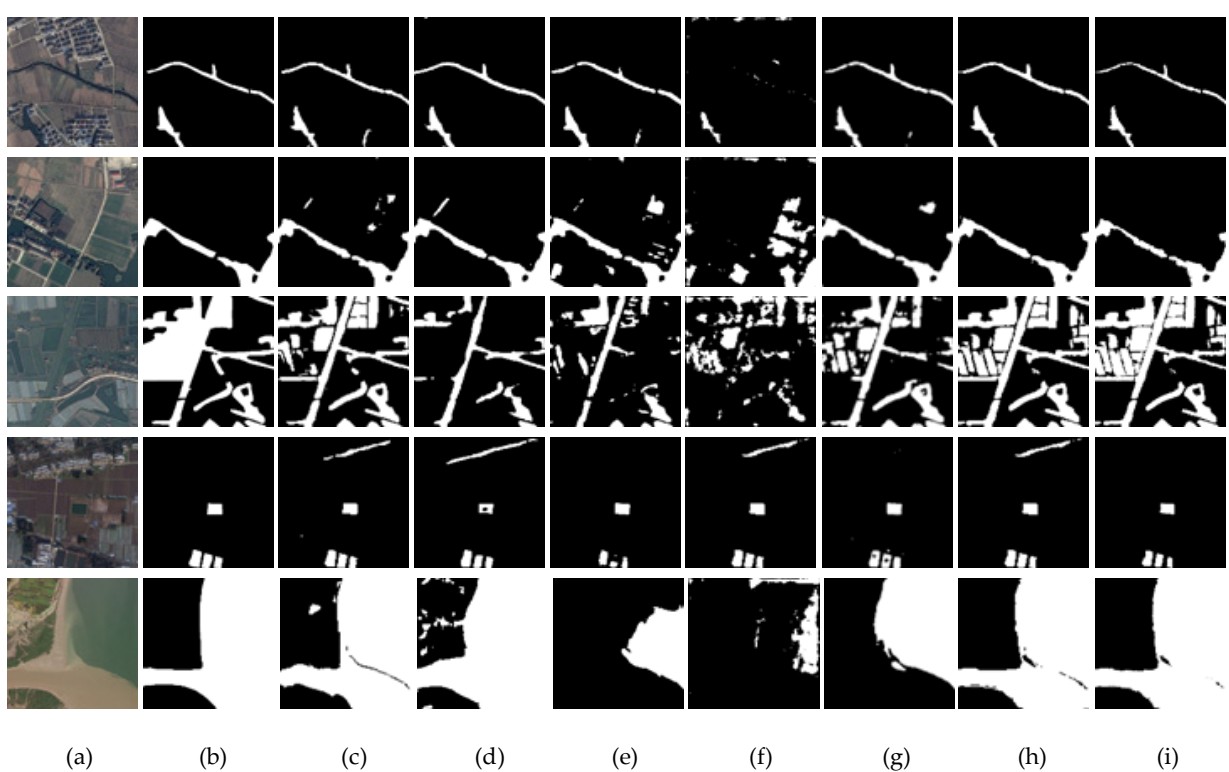

(a)     (b)     (c)     (d)     (e)     (f)     (g)     (h)     (i)

**Figure 5.** Qualitative comparisons with the other state-of-the-art methods in the source domain. (**a**) Image; (**b**) Ground-truth; (**c**) HRnet V2; (**d**) Bisenet; (**e**) U-Net; (**f**) DFAnet; (**g**) DeepLab V3+; (**h**) SFnet; (**i**) SFnet + CRF.

### 3.3.2. The Comparison of Unsupervised Methods

To evaluate the performance of SFnet-DA in segmenting water bodies, we use the combination of DeepGlobe Land Cover Classification Challenge and 2020 Gaofen Challenge dataset as the source domain of the network, and only use images of the WHDLD as the target domain of the network. In order to test the effect of the compared unsupervised methods, we try to classify one category into water and background, respectively, and select a classification method with high accuracy. The predictions of ISODATA [41] and ISB [42]

are not the binary tasks and the number of predicted categories are not fixed, which is not conductive to our qualitative measurement. Therefore, in the quantitative comparisons, we only compare UIS, K-means and AC algorithms. The comparisons between our method and the current excellent methods are shown in Table 2. Compared with the other unsupervised methods, our method shows great advantages. Among them, *MIOU* is 33.17% higher than UIS, 34.23% higher than K-means and 30.79% higher than AC. Table 2 also shows the time required for each network to train an image. Our speed is slower than K-means, which is normal because we use the DL method.

**Table 2.** Quantitative comparisons of models in the target domain.

| Method | PA | $F_1$ | MIOU | Time (s) |
|---|---|---|---|---|
| K-means [38] | 72.91 | 77.7 | 49.76 | 0.09 |
| UIS [39] | 54.94 | 79.25 | 50.82 | 0.38 |
| AC [40] | 78.15 | 79.03 | 53.2 | 0.74 |
| Ours | 88.77 | 92.56 | 83.99 | 0.2 |

The predicted charts of ISODATA and ISB are processed in conjunction with the water bodies in the labels, with the part of the water body set to white and the others black. As shown in Figure 6, compared with other unsupervised methods, our method is closest to the ground truth. K-means is easily disturbed by green plants and bright objects, resulting in false detections. The detection effect of ISODATA is better than other compared methods, but it depends on texture features and would produce false detections. In addition, because the compared methods fail to make full use of image features, the edge of predicted images is blurred, and there is a lot of noise which results in the discontinuity of water bodies. By using the training parameters of existing datasets, our method makes full use of the powerful feature extraction ability of DL, so the detection accuracy is higher, and the water edge is clearer.

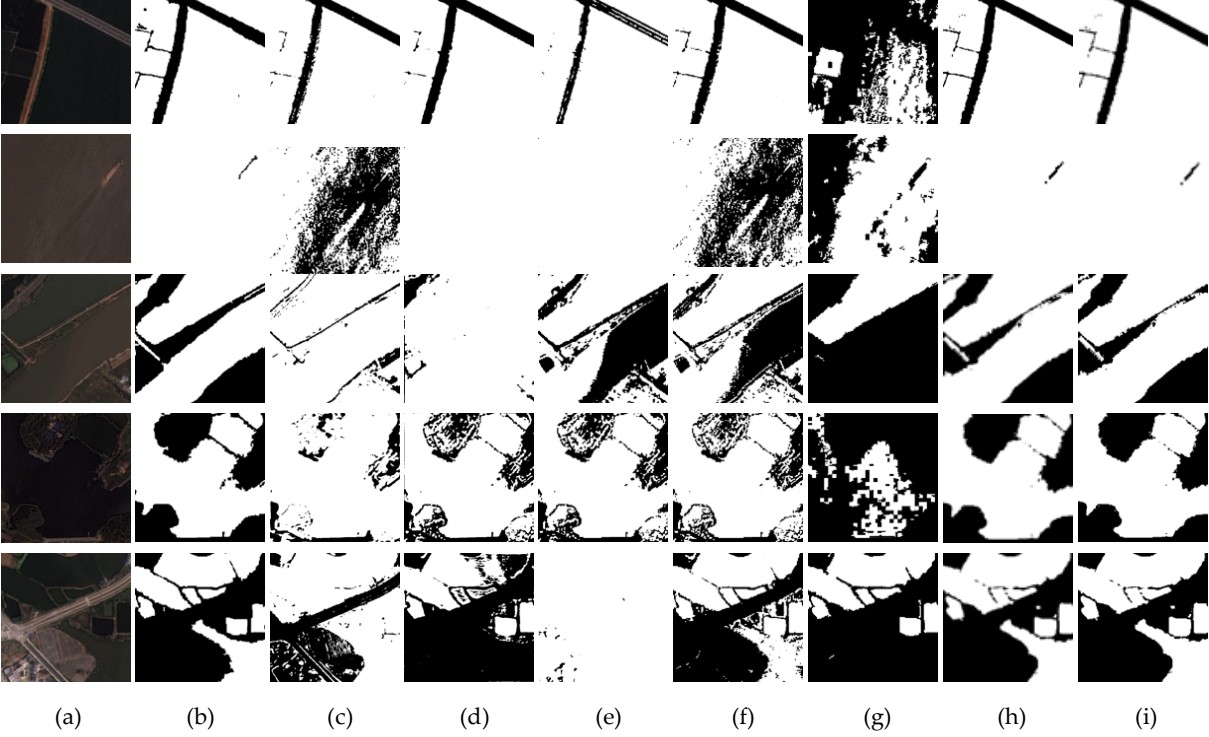

**Figure 6.** Qualitative comparisons with the other in the target domain. (**a**) Image; (**b**) Ground-truth; (**c**) K-means; (**d**) AC; (**e**) UIS; (**f**) ISODATA; (**g**) ISB; (**h**) SFnet-DA; (**i**) SFnet-DA + CRF.

Therefore, compared with other methods, our approach can take advantage of existing labeled datasets and migrate them well to water detection in unlabeled remote sensing datasets.

## 4. Discussion

Our research aims at the automatic extraction of water information from remote sensing images, and the experimental results show that our method is better than other compared methods. From the experimental results in Section 3.3.1, it can be found that Bisenet and HRnet V2 have better detections for small rivers because they retain shallow information, but the detection effects are poor in complex scenes. Deeplab V3+ uses dilated convolution to obtain context information, which makes it have a good effect in complex backgrounds, but Deeplab V3+ has missed detection for small rivers. Noting the phenomenon that shallow information is helpful for detailed information and deep information is helpful for semantic information, our method adopts the SSA module to selectively enhance shallow or deep information and uses MFF to refine the edge of the water body. The quantitative and visual results show that our method has good performance in the source domain. To further test the advantages of our network, ablation experiments are conducted here. We sequentially deleted the fully connected CRFs, MFF module and SSA mechanism in the network, and compared them with the backbone network HRnet V2 under the same source domain dataset. The experimental results are shown in Table 3. It can be found that SSA improves *MIOU* by 1.4%, MFF by 0.42%, and CRF by 0.42%. To further evaluate the SSA mechanism, we visualize the output images before and after the SSA mechanism in the form of a heat map, as shown in Figure 7. It can be found that the network after adding SSA can efficaciously resist the interference of noises such as the shadow, buildings, green algae, and vegetation, reducing the false or missed detections. Additionally, the network after joining SSA has better identification effects and robustness for vast lakes and small water regions. Comparing the images in Figure 8, it can be seen that the network can better obtain the edge information of the images after adding MFF. In addition, from the experimental results in Section 3.3.2, it can be found that the effects of simple unsupervised detections are not ideal, while our method has been greatly improved compared with them because of the powerful feature extraction ability of DL. From Tables 1 and 2, it can be found that in the source domain, our method is faster than some networks but slower than efficient networks, and in the target domain, it is faster than other methods but slower than K-means, which needs to be improved in the future.

**Table 3.** Evaluation of water body extraction results obtained by different models.

| Method | PA | F$_1$ | MIOU |
|---|---|---|---|
| HRnet V2 [20] | 88 | 97 | 84.35 |
| HRnet V2 + SSA | 89 | 97 | 85.75 |
| HRnet V2 + SSA + MFF | 89 | 97 | 86.17 |
| HRnet V2 + SSA + MFF + CRF | 90 | 97 | 86.59 |

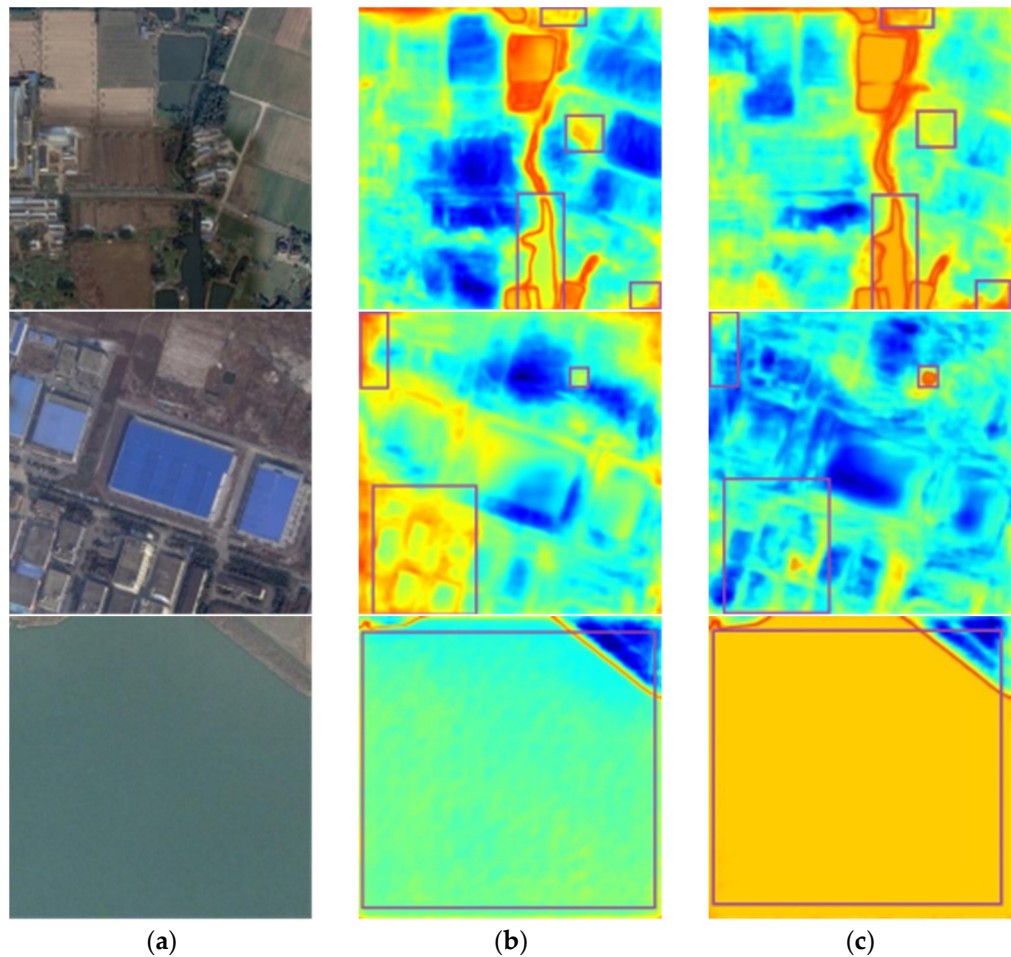

**Figure 7.** Visual comparison results of the SSA mechanism. (**a**) Image; (**b**) without the SSA mechanism; (**c**) with the SSA mechanism. The boxes circle the changes after adding SSA.

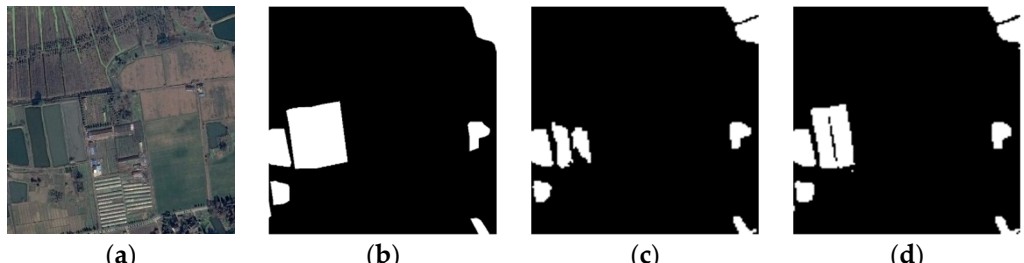

**Figure 8.** Results of ablation experiment. (**a**) Image. (**b**) Ground truth. (**c**) HRnet V2 + SSA. (**d**) HRnet V2 + SSA + MFF.

## 5. Conclusions

The automatic extraction of water resource information plays an important role in water resource management. In addition, remote sensing images that can easily obtain fine spatial and temporal resolution information are increasingly used for large-scale water extraction. Many methods have been developed and have shown good results in extracting various types of water bodies, including rivers, lakes, and coastlines. However, there are still some areas for improvement. Water index methods, such as AWEI and WI$_{2015}$, are widely used but perform poorly in some areas, and machine learning methods, such as SVM and J48 decision tree, depend on specific environments and the selection of training data. Deep learning methods have powerful extraction capabilities and show considerable potential, but they lead to loss of resolution and rely on large amounts of data. In this paper,

a novel SFnet-DA network is proposed to accurately extract water bodies from remote sensing images, which is a combined network of SFnet and domain adaptation (DA). Benefits from SSA mechanism, SFnet-DA can selectively extract water bodies of different sizes and alleviate the influence of noise. To ensure the edge information of water body, the SFnet-DA embeds MFF module and uses full connection CRFs to optimize the prediction image. Based on the domain adaptation, SFnet-DA makes full use of existing datasets to predict unlabeled images. To verify our method, the DeepGlobe Land Cover Classification Challenge and 2020 Gaofen Challenge dataset are combined into the source domain, and the quantitative and qualitative results are superior to other methods. In the target domain, the results demonstrate the SFnet-DA is also much greater than other methods in the Wuhan dense labeling dataset (WHDLD). However, it can be found that the accuracy of the target domain is lower than that of the source domain and the network efficiency is lower than some methods. In future work, we will focus on how to further reduce the feature difference between the source domain and the target domain and improve the efficiency of the network.

**Author Contributions:** J.L. supervised the study, designed the architecture, and revised the manuscript; Y.W. wrote the manuscript and designed the comparative experiments. All authors have read and agreed to the published version of the manuscript.

**Funding:** This work was supported in part by the Innovative Talent Program of Jiangsu under Grant JSSCR2021501, by the China High-Resolution Earth Observation System Program under Grant 41-Y30F07-9001-20/22, and by the High-Level Talent Plan of NUAA, China.

**Data Availability Statement:** Not applicable.

**Acknowledgments:** The authors would like to thank the editors and the reviewers for their valuable suggestions.

**Conflicts of Interest:** The authors declare no conflict of interest.

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
