# Peer review of "Water Body Extraction in Remote Sensing Imagery Using Domain Adaptation-Based Network Embedding Selective Self-Attention and Multi-Scale Feature Fusion"

_remotesensing, doi:10.3390/rs14153538_

Round 1

Reviewer 1 Report

This is an interesting article about the advanced use of deep learning techniques for mapping of water on high resolution RGB satellite images. The focus is clearly on deep learning, and not so much on remote sensing, therefore it is not sure that the Journal of Remote Sensing is the best place for this otherwise high-quality research. I only have a few minor remarks:

-          The English is good, understandable, but there are quite some style and grammar errors (e.g. in many places “image” is used instead of “images”). I suggest a thorough review of the English

-          The introduction provides an overview of earlier research related to water mapping using satellite imagery and the use of deep learning in remote sensing. Especially, the overview of water mapping is rather limited. Lots of research has been published in this field and the authors present only very view. Also, the literature overview of deep learning approaches in RS is superficial. Both types of literature research are almost completely limited to studies conducted by Chinese authors. A broader perspective is advised.

-          Line 264-269. This section is more appropriate in the results/discussion section

-          Figure 7. It would be good to present the results of the DeepLab model next to the results of the authors, since after SFnet+CRF DeepLab provides the best results. This would make it easier to compare the two results visually.

Author Response

Please check the respones in the supplementary.

Reviewer 2 Report

The overall structure of the article is reasonable, the logic is clear, and the research field is relatively novel, but there are still some detailed problems that need to be modified: 1 The overall structure of the abstract is not clear enough. A brief description of the research background should be added at the beginning. The research methods need to be more detailed and the research results should be simplified. 2 The second part of the introduction needs to summarize the existing research, and the last paragraph should highlight the research methods, research contributions, and innovations. 3 Explanation of Figure 3 and Figure 5 is not clear enough these should be supplemented to make the connection clearer. 4 There is a problem with the title number of some sections. 5 Figures are too small to see clearly. 6 At the beginning of the conclusion part, the description of the research background needs to be added. In addition, the research process needs to be simplified to increase the deficiencies in the research and the future research direction.

Author Response

Please check the response in supplementary.

Reviewer 3 Report

I found this article interesting and valuable to be published. However, I think this comment needs to be taken into account before it has to be accepted.

Please add more detail regarding the methods and perhaps a research methodology that shows the main step steps of the research will be quite helpful.

the results and discussion sections ar mixed up,  which I think it is not a good idea while the authur are supposed to give more details regarding the method and its efficiency n the discussion section that is missing now. 

Please add the state of art and the contribution of research as advanced and progressive research in remote sensing to the conclusion section. 

Author Response

(The authors gave the same response as above.)

Reviewer 4 Report

In this paper, the authors present "Water Body Extraction in Remote Sensing Imagery Using Domain Adaptation-Based Network Embedding Selective Self-At-tention and Multi-Scale Feature Fusion."  The MFF module is used to accurately acquire edge information by changing the channel number of advanced feature branches with a unique fusion method. To skip the labeling work, SFnet-DA reduces the difference in feature distribution between labeled and unlabeled datasets by building an adversarial relationship between the feature extractor and the domain classifier, so that the trained parameters of the labeled datasets can be directly used to predict the unlabeled images. Experimental results demonstrate that the proposed SFnet-DA has better performance on water body segmentation compared with some other methods published in recent literature. However, there are some issues should be addressed.

1. For the water segmentation of high-resolution remote sensing images, the accurate location of water-body is instrumental in correct positioning and area calculation. How to reduce the number of feature channels? 

2. The water detection methods based on deep convolution neural network utilize a large array of labeled samples for training. However, labeling is extremely time-consuming and expensive. How to reduce time-consuming?

3. The model trained on one labeled dataset cannot predict another unlabeled dataset since there are huge differences, such as shooting area, image resolution. How to solve the problem? 

4. Conditional random field (CRFs) are widely utilized to smooth rough segmented images. However, CRFs only use a single potential energy on a pixel or paired potential energies on adjacent pixels. This property results in a sparse structure that can only couple adjacent points and implement modeling between short-distance pixels. In contrast to CRFs, each pixel in fully connected CRFs is coupled with all other pixels to form connecting edges, which greatly enlarges the quantity of paired connections and is conducive to establish long-distance connections, restore edges, and identify small water bodies. How to perform the proposed network with CRFs?

5. In Figure 7, all of the images in row 7, including the images predicted by SFnet, are affected by bushes. Although U-Net and Deeeplab v3 + obtain the context information without false detection, there are some missing detections. How to solve this problem?

6. How to overcome the defects using selective feature enhancement and multi-scale feature fusion network based on domain adaptation (SFnet-DA) in this work?

7. To the different datasets, the differences of water bodies are more common and lead to the domain shift, namely dataset bias. How to overcome this problem?

8. How to catch precise segmentation for unlabeled datasets in this paper?

Author Response

(The authors gave the same response as above.)

Reviewer 5 Report

The submitted manuscript proposed a method for water body identification using neural networks, domain adaptation, self-attention and multi-scale fusion concepts. The manuscript is well-written and the experiments show interesting results. However, some minor adjustments are required. Please, see the comments and questions below:

Comments:

  • The definition of SFnet-DA is missing in the abstract;
  • Define the DP_CONV elements in Figure 3;
  • Formal definitions for W and b_att are missing (Eqs. 3~5);
  • The “evaluation metrics” section is unnecessary – it suffices to include only references for Eqs. 22~25;
  • Include more details about the adopted computational platform (i.e., computer configuration, programming language and libraries);
  • Include the computational time for training the proposed method and its competitors;
  • Revise the title of Section 3.5.2 – the methods used for comparison are not “networks.” Moreover, unsupervised methods like K-Means and ISODATA produce groups (of similar objects) where it is not necessarily possible to distinguish water from other classes. In this context, how the obtained clusters allowed to identify water/non-water areas?
  • It is highly recommended to make the code available in a public repository.

Author Response

(The authors gave the same response as above.)

Round 2

Reviewer 3 Report

thanks for the comperhanive revise 

Reviewer 4 Report

The authors have solved the related problems.